# Characterising people who inject drugs, and association with HIV infection: A situation analysis in Kampala City, Uganda

Peter Mudiope[1]*, Bradley Mathers[2,3], Joanita Nangendo[4], Samuel Mutyaba[1], Byamah B. Mutamba[5], Stella Alamo[6], Nicholus Nanyenya[1], Fredrick Makumbi[1], Miriam Laker-Oketta[7], Rhoda Wanyenze[1]

**1** Makerere University School of Public Health, Kampala Uganda, **2** Global HIV, Hepatitis and Sexually Transmitted Infections Programs, World Health Organisation, Geneva, Switzerland, **3** Kirby Institute, University of New South Wales Sydney, Sydney, Australia, **4** School of Medicine, Makerere University College of Health Sciences, Kampala, Uganda, **5** Butabika National Referral Mental Hospital, Kampala, Uganda, **6** United States of America Centers for Disease Control and Prevention, Kampala, Uganda, **7** Infectious Diseases Institute, Makerere University College of Health Sciences, Kampala, Uganda

* pmudiope@gmail.com

## Abstract

Uganda implements interventions for injection drug use, but significant barriers hinder efforts to effectively reach and support persons who inject drugs (PWID). We describe characteristics of PWID, and associated risk behaviour, to inform the designing of programmes that are tailored to clients' needs and preferences. A cross-sectional survey (August 23rd to December 5th, 2023) in Kampala interviewed 354 PWID (≥18-years) at selected venues(bars,lodges, street corners and ghetto). Peer eductors and counsellors administered a structured questionnaire covering socio-demographics, drug use, sexual risk, and medical history. HIV serostatus was determined by self-report or testing for consenting participants without history of recent testing Binary logistic regression was used to establish the relationship between HIV infection and risky drug- and sexual behaviour of PWID. Participants were predominantly Ugandan (95.2%), male (73.2%), unmarried (55.9%), unemployed (81.8%), with higher levels of education and varying ages. Mental disorders were prevalent, with 48.7% reporting at least one underlying condition, including depression (30.8%) and anxiety (9.6%). Physical health issues included fever (32.9%), cough (32.5%), malaria (22%), and sexually transmitted infections (15%). Over 82.6% were introduced to drugs by close acquaintances. HIV prevalence among participants was 3.7%, higher in females (8.4%) and non-Ugandans (16.7%). Being female and experiencing difficulty accessing sterile injection materials were associated with HIV-positive status. Our study provides valuable insight into the socio-demographic, mental, physical health, and HIV risk behaviour of PWID in Kampala, Uganda. The findings indicate significant vulnerabilities to injecting drug use, mental disorders, and high-risk behaviors that predispose this population to HIV infection. Despite a low HIV prevalence in this population compared to previous estimates, the interplay between drug use, risky injecting practices, and sexual behaviour suggests an urgent need for targeted interventions to address these intertwined challenges.

**Data availability statement:** All data can be found in the manuscript and supporting information files.

**Funding:** Research reported in this publication was supported by the Fogarty International Center and the United States of America National Institute on Mental Health of the National Institutes of Health under Award Number D43 TW010037. The content is solely the responsibility of the authors and does not necessarily represent the official views of the National Institutes of Health.

**Competing interests:** The authors declare that they have no competing interests.

**Abbreviations:** CHAU, Community Health Alliance Uganda; IDU, Injecting drug user; MOUD, medication for opioid use disorder; OUD, Opioid use disorder; PWID, People Who inject drugs; UHRN, Uganda Harm Reduction Network Organisation.

## Introduction

Recent global data indicate an increasing supply and demand for illicit drugs. The United Nations Office for Drugs and Crime (UNODC) reports that illicit drug use increased by 23% from 240 million in 2011 to 296 million in 2021, of whom 13.2 million people were injecting drugs[1]. Louisa Degenhardt et al (2023) estimated that 14·8 million people are drug injectors in the 190 countries that contribute more than 99% of the global population[2]. The rising burden of illicit drug use including injecting and challenges related to its regulation, continues to harm the economic, health, and social sectors [1].

The high prevalence of HIV and Hepatitis B and C infections in sub-Saharan Africa has in part been attributed to thesobserved increase in injection drug use [3–5]. This problem is compounded by a high occurrence of infectious diseases, as well as a myriad of socio-demographic, mental health, and other general physical health issues among PWID. Studies have reported injection drug users to be mainly men, not married, unemployed, often young with varied education levels, and prone to high risk of sexual and social crime [3]. Concurrent mental disorders with Opioid use disorder (OUD) also present a dual burden for PWID. A high prevalence of depression and anxiety among people with OUD was reported in sub-Saharan Africa [6]. However the prevalence of mental disorders among PWID varies widely based on the method of ascertainment [7–9]. A recent study indicated that the prevalence of depressive disorders varied from as low as 3·44% to a high 66% in the general population [2].

In Uganda, an estimated 7,356 people are injecting drug users, with an HIV prevalence of 16% compared to 5.6% in the general adult population. Hence injecting drugs is the second highest contributor to new infections among key populations in the country[10]. Critical attention is needed to reduce HIV transmission among PWID and other key populations. Uganda adopted the WHO/UNODC recommended HIV prevention interventions for PWID as outlined in the national strategic plan [11, 12]. Working with the Ministry of Health, the Uganda Harm Reduction Network (UHRN), the Global Fund and UN health development partners supported a project that distributed clean needles and syringes in Kampala and Mbale cities [13]. In 2020, with PEPFAR support, Uganda set up the first medication for opioid use disorder (MOUD) clinic at a psychiatric hospital in Kampala city

Hitherto another clinic, has been established in Mbale city in the eastern part of the country. Despite these efforts, the scope of service and geographical reach of HIV prevention, care, and treatment services for PWID is far from realising the targeted 90%. The barriers to optimally reaching PWID with such services have previously been described to act at multiple levels including social, physical, economic or political [14, 15]. Key barriers include; stigma, peer influence, criminalising of drug use, lack of transport to the service points, comorbidities such as mental disorders, limited knowledge of available services, and limited capacity to provide services[14, 15].

There is a need to improve service delivery by offering differentiated harm reduction services based on client needs and preferences. This requires understanding the risk profiles of PWID regarding their socio-demographics, drug use, risk behaviour, and criminal and mental health among others. PWID risk profiles are heterogeneous and given the limited resources, tailored services based on PWID needs and preferences are required [16]. In Uganda, evidence on the relationship between PWID' characteristics and their risk of HIV infection is still limited. Earlier studies were primarily exploratory with small sample sizes and did not fully captured the variations in participants' characteristics related to their risk of HIV infection and access to HIV prevention services [17]. Knowledge of the varying profiles of PWID is crucial for the implementation and uptake of risk reduction interventions. This study aimed to describe the characteristics of PWID, physical and mental health, and HIV risk behaviour to inform the design of programs that are tailored to client's needs and preferences.

## Methods

### Study design & setting

This cross-sectional survey was implemented from August 23rd 2023 to December 5th 2023 at selected venues (bars, lodges, street corners and ghettos) in Kampala City. The study was conducted with the active involvement of the Uganda Harm Reduction Network (UHRN), a PWID community-based organization that provides harm reduction services to PWID in the community.

### Study population, sampling, and sample size

In total, 38 venues (bars, lodges, street corners and ghettos) frequented by injecting drug users were conveniently selected based on the safety, security status, and willingness of local leaders to permit data collection. We recruited 354 injecting drug users above 18 years who had a history of at least one drug injection in the previous three months, as established by history and the manifestation of recent skin injection marks. Recruitment within the community utilised peer educators who mobilized participants to participate in the study. Interviewed participants were encouraged to refer other users within their network, helping the study to achieve an adequate sample size.

### Data collection

The data were collected using a structured questionnaire and keyed directly into open data kit (ODK) database software[18]. The questionnaire was piloted among PWID attending the MOUD clinic at Butabika National Referral Mental Hospital. The MOUD clinic staff were trained to collect data. Two additional peers were recruited and trained to mobilize venue owners and participants to support and participate in the study respectively. During visits to the hotspot or drop-in centers, the data collectors administered the questionnaire in the participant's preferred language. The data were entered directly into the ODK, where the data manager provided the quality control feedback for redress by data collectors.

To increase the likelihood of participation, we leveraged existing social networks, encouraging initial participants who would be willing to participate in the study. We prioritised flexible scheduling of interviews to accommodate the participant's availability while considering their mental status, work schedules, treatment appointments, and other commitments. Due to legal restrictions and the potential risk of arrest, accessing PWID posed significant challenges [19]. Therefore, we prioritised the safety and well-being of both participants and data collectors. Throughout the data collection process, participants were assured of their safety and the confidentiality of their information. They were also informed about the importance of their participation in the study for informing public health interventions.

### Study measurements

Participants' sociodemographic characteristics included age, sex assigned at birth, religion, nationality, education level, monthly income, and source. Regarding substance use, participants were asked which substances they often used/injected in the previous 30 days, the lifetime duration of use, reasons for use/injection, and the frequency of use/injection. Questions on the frequency and use of flashblood, sharing needles/syringes, and frequency of experiencing overdose were asked to establish the injection risk behaviour. Sexual risk behaviour was assessed by asking about the sexual encounters (vaginal or anal) with non-regular partners and condom use in the last 30 days. Participants were considered to have consistently used condoms if they used condoms for all times during all the sexual encounters. Sexual risk was

also assessed by asking participants if they had sex with someone else in exchange for money, drugs, or any other commodity as the term for payment. Physical illness in the past three months and screening for HIV, TB, and hepatitis were also assessed. Those who reported being HIV positive and had their status confirmed from clinical records were not required to undergo additional testing. For others, HIV status was determined using the Determine™ HIV-1/2 rapid test, with confirmation via the Bioline HIV1/2 3.0 rapid test, following national guidelines [20]. The state of mental illness was assessed through self-reporting, asking if particpants were or had been on treatment for a mental health condition. Sexual gender-based violence (SGBV) was evaluated by asking if a participant had ever been forced to have sexual intercourse. Other factors assessed included the history and frequency of incarceration.

## Statistical analysis

Baseline socio-demographic, mental, and physical health-related, drug use behaviors and sexual risk profiles were summarised using medians and interquartile ranges for numerical variables and proportions for categorical variables. Binary logistic regression, was used to determine the association between the independent variables (socio-demographic, drug use, and sexual-related risk profiles) and the HIV status of participants.

During bivariate analyses, variables that had p-values <0.1 were selected for inclusion into multivariate analysis. In multivariate logistic analysis level, we adjusted for confounding using the backward stepwise method based on p-values, removing most non-significant variables and interaction terms from the model until only significant variables remained. The variables with p-values <0.05 were reported as significantly associated with HIV status. Adjusted Odds ratios (AORs) and the corresponding 95% confidence intervals (CI) were reported. All analyses were performed using the STATA version. 14.2 (College Station, Texas).

## Ethical considerations

This study was approved by the Makerere University School of Public Health Higher Degrees Research and Ethics Committee (Protocol number SPH-2021-166), and the administration of Butabika Hospital. All the participants were interviewed after providing written informed consent.

## Inclusivity in global research

Collaborating with UHRN staff, venue owners, and peer educators helped to reassure participants about the confidentiality of their involvement and alleviate concerns about potential legal repercussions. Additional information regarding the ethical, cultural, and scientific considerations specific to inclusivity in global research is included in the supporting information (S1 Checklist).

## Results

A total of 354 predominantly male (73.2%) participants were interviewed. They were of Ugandan nationality (94.9%), unmarried (52.8%), and not in regular employment (82.2%). The median age was 32 years, interquartile range (IQR), (26-37). A significant percentage of participants resided either alone (49.7%) or with a partner or friend (45.5%), identified with the Anglican/Born-Again faith (41.5%) or Roman Catholic(22%) or Muslim(34.5%), and had achieved at least a secondary level of education (44.9%). A considerable proportion (59.6%) reported a monthly income of between 50,000/= and 500,000/= Ugandan shillings, while the majority (74.8%) disclosed a history of prior detention or incarceration (Table 1).

## Mental health illnesses

In total, 136 individuals (38.4%) disclosed having at least one underlying mental illness, among whom 27 (19.8%) were undergoing treatment, and 17(12.7%) reported more than one existing mental condition. Among those who reported mental illness, 43 (12.1%)

**Table 1. Socio-demographic characteristics of people who use illicit drugs in Kampala, Uganda (N = 354).**

| Variable | Number(Percentage) |
|---|---|
| **Sex** Male n (%) | 259 (73.2) |
| Female n (%) | 95 (26.8) |
| **Age** -median (interquartile range-IQR) | 32(26 -37) |
| 18-24yrs | 56 (15.8) |
| 25-34yrs | 187 (52.8) |
| 34yrs & above | 111 (31.4) |
| **Highest education level attained(n=424)** | |
| No education | 47 (13.3) |
| Primary level | 96 (27.1) |
| Secondary | 159 (44.9) |
| Post-secondary | 52 (14.7) |
| **Married Status** | |
| Married | 15 (4.2) |
| Cohabiting | 43 (12.2) |
| Separated/widowed | 109 (30.8) |
| Un-married | 187 (52.8) |
| **Religion** | |
| Anglican/Born-Again | 147 (41.5) |
| Roman Catholic | 78 (22) |
| Islam | 122 (34.5) |
| No religion | 7 (2) |
| **Nationality** | |
| Ugandan | 336 (94.9) |
| Non-Ugandan | 18 (5.1) |
| **Stay with** | |
| Alone | 176 (49.7) |
| Parent/children | 17 (4.8) |
| Partner/friend/relative | 164 (45.5) |
| **Employment status(source of income)** | |
| Regular employment | 63 (17.8) |
| Informal casual work | 155 (43.8) |
| Other activities(such as sex work) | 96 (27.1) |
| No source of income | 8 (2.3) |
| **Monthly Income(UGX)** | |
| <=50,000/= | 83(23.5) |
| >50,000-500,000ugx | 211 (59.6) |
| >500,000/= | 60 (16.9) |
| **Ever been imprisoned for more than 24 hours in life** | |
| Yes | 265 (74.8) |
| No | 87 (24.6) |
| Decline to respond | 2 (0.6) |

acknowledged co-existing anxiety, with five of them receiving treatment; 112 (31.6%) indicated co-existing depression, with 25(7.1%) undergoing treatment. Furthermore, out of the 26(7.3%) individuals who reported co-existing Attention Deficit Hyperactivity Disorder (ADHD), two were receiving treatment. Other reported mental illnesses included schizophrenia (3 cases), bipolar disorder (1 case), and personality disorder (4 cases) (Fig 1).

## Physical health illnesses

In terms of physical health, a total of 129(36.4%) participants reported experiencing fever within the three months preceding the interview, while 127(35.9%) mentioned having a cough, 15(4.2%) reported frequent urination, 28(7.9%) experienced painful urination, and 7(2%) had groin wounds. Additionally, 82 (23.2%) individuals had been treated for malaria, 9(2.5%) for sexually transmitted infections (STIs), 14 (3.9%) for urinary tract infections (UTIs), and 22(6.2%) for upper respiratory tract infections (URTIs). Five participants had received treatment for tuberculosis, and 20(4%) for skin diseases.

## Drug use-related characteristics

The median duration of drug use was 8.1(IQR: 4.1-13.8) years. 412 (82.6%) participants, revealed that they were introduced to drug use by close friends or relatives, with 72.5% citing peer influence as the primary reason for continued drug use. Other reported reasons included coping with the loss of a loved one in 57 individuals (11.4%), coping with work-related stress(six participants), and personal illness (six) as reasons for continued drug use. About 348(98.5%) participantshad used heroin in the last three months, of whom 28 (8.0%) were daily users. Of the 153 participants who used cocaine (crack) in the last three months, 110 (64.7%) were daily users. The number of participants who used other substances in the last three months included 266 who used marijuana, 225 who consumed alcohol, and 206 who used tobacco (Fig 2).

Among the participants, 178 (50.3%) had last injected more than a month before the interview, 62 (17.5%) within the last month, 61 (17.2%) within the last week, and 21 (5.9%) on the day of the interview. Regarding injection frequency, most participants 199 (57.2%) reported occasional injections, 54 (15.5%) injected at least monthly, 67 (19.3%) injected weekly, and 28 (8%) injected daily. Of the 55 participants who reported being on MOUD, 35 injected drugs only occasionally (Table 2).

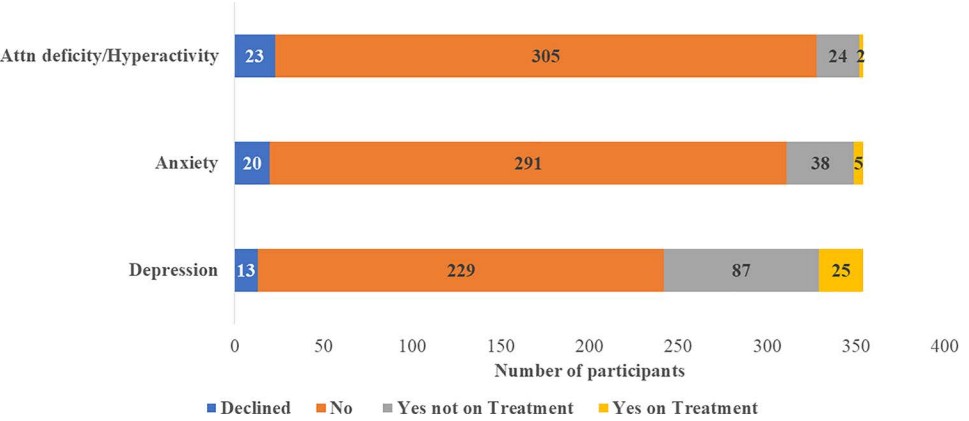

**Fig 1. Mental Health Illnesses(self-reported) among people who inject drugs in Uganda (n =354).**

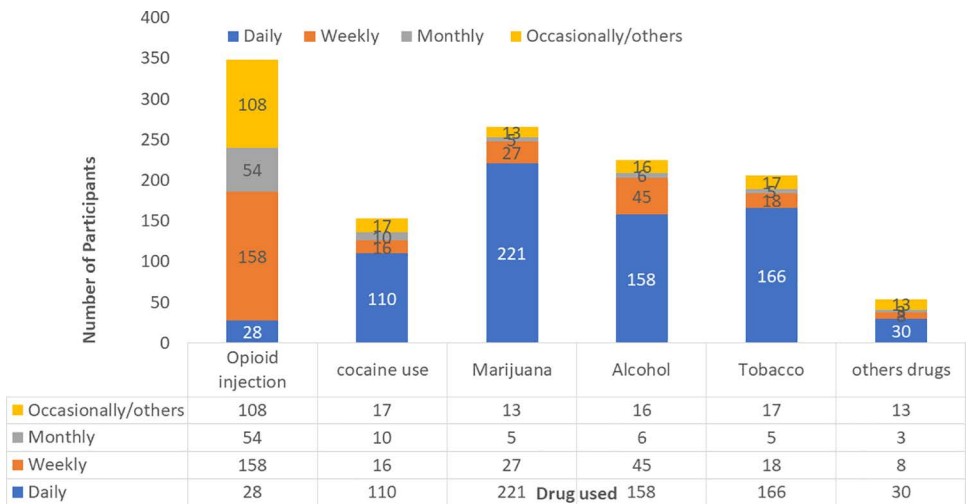

**Fig 2. Frequency of commonly used drugs among drug users in Uganda (n = 354).**

## HIV Risk behaviour

Overall, 9.3% reported using a shared needle or syringe, and a similar proportion acknowledged the sharing of other injecting materials (water, tourniquet, spoons, and filters). Additionally, 6.2% reported a risky practice of using flashblood, and using unsterilized syringes and needles, either on themselves or another person. The majority of participants (75.7%) stated that obtaining clean needles and syringes was easy (Table 3).

## Drug use and sexual-related HIV risk factors

Regarding sexual behaviour, 53.4% participants reported engaging in sexual encounters with non-regular partners in the last three months, of whom 19.8% reported that they knew the HIV status of their non-regular partners. Only 32.6% and 25.2% of participants reported consistent condom use during sex with a non-regular partner, and the last three sexual encounters, respectively. Over 128(67.4%) participants reported engaging in paid or paid-for sex encounters, and 45.3% exchanging sex for drugs. Additionally, 53.9% of participants reported using drugs before engaging in sexual encounters with other partners (Table 3).

## HIV positivity among people who inject drugs

Overall, HIV status was ascertained in the 354 drug injecting participants, with 13 (3.7%) HIV positive. HIV positivity was higher among females (8.4%) compared to males at 1.9%. A higher proportion of younger persons (18-24 years) were HIV-positive (7.1%) compared to older aged 25-34 years (2.1%) and those above 34 years (4.5%). The proportion of PLHIV was higher (16.7%) among non-Ugandans compared to Ugandan participants (3%). There was no variation in positivity based on participants' marital status and income.

Regarding HIV risk behaviour related to drug use, HIV was more prevalent among participants who reported difficulty in accessing clean injection materials. About 5.3% of those with difficulty in accessin clean injection materials compared to 1% of those with ease of accessing clean injection materials were HIV positive. The reported proportions of PLHIV did not vary based on duration of injection, and frequency of injecting.

In terms of sexual risk behaviour, HIV occurrence was higher among participants who reported engaging in sexual activities with non-regular partners. No variation in HIV status

**Table 2. Drug use-related characteristics of persons who inject illicit drugs in Kampala, Uganda (N = 354).**

| Variable | Number (%) |
| --- | --- |
| **Drug use duration median(IQR) years** | 8.1 (4.1-13.8) |
| Less than 5 years | 112 (31.6) |
| 5 to 10 years | 98(27.7) |
| Above 10 years | 144(40.7) |
| **Reasons for drug use** | |
| Cope with the loss of a loved one | 60(16.9) |
| Cope with stress at work | 42(11.9) |
| Peer Influence | 246(69.5) |
| Cope with personal illness | 6(1.7) |
| **Who introduced you to drug use?** | |
| Drug dealer | 18(5.1) |
| Friend/relative | 287(81.1) |
| Oneself | 49(13.8) |
| **Experience drug overdose last year** | |
| Yes | 73(20.6) |
| No, or declined to Answer | 281(79.4) |
| **Drug injection duration median (range) Years** | 1.8 (0.1-24.6)years |
| **Frequency of drug injecting** | |
| Daily | 28(8.0) |
| Weekly | 67(19.3) |
| Monthly | 54(15.5) |
| Occasionally | 199(57.2) |
| Declined to answer | 6(1.7) |
| Ever enrolled in the MOUD programme | |
| Yes | 68(19.2) |
| No | 286(80.8) |
| Ever enrolled in the NSP* | |
| Yes | 72(20.3) |
| No | 282(79.7) |
| Ever admitted to a rehab/detox program | |
| Yes | 75(21.2) |
| No | 279(78.8) |
| Ever treated in other ways (OPD prayer/tradition) | |
| Yes | 86(24.3) |
| No | 268(75.7) |

*NSP refers to needle syringe exchange programme

was reported based on other sexual risk behaviour such as consistent condom use, awareness of partner's HIV status, and transactional sex (paying money or other goods for sex, and taking drugs for sex) (Table 3).

## Association between socio-demographics, risk behaviors, and HIV

At bivariate analysis, female sex, non-Ugandan by nationality, with difficulty in accessing sterile injection materials, and a history of taking drugs before sex, were more likely to be HIV positive. The age of the participant, having sex with a non-regular partner in the last three

**Table 3. Drug use and sexual-related HIV risk factors among injection drug users in Kampala, Uganda (n = 354).**

| Variable | number (%) |
|---|---|
| **Needle sharing in the last six months** | |
| Yes | 33(9.3) |
| No | 277(78.3) |
| Declined to answer | 44(12.4) |
| **How easy to get sterile injection materials** | |
| Easy | 268(75.7) |
| Not easy | 41(11.6) |
| Declined to answer | 45(12.7) |
| **Ever used flash blood** | |
| Yes | 22(6.2) |
| No | 318(89.8) |
| Declined to answer | 14(3.9) |
| **Ever shared other injection equipment in last 6months** | |
| Yes | 33(9.4) |
| No | 272(76.8) |
| Declined to answer | 49(13.8) |
| **Sex with a non-regular partner(co-current sex partner)** | |
| Yes | 189(53.4) |
| No | 165(46.6) |
| **Know Partner HIV Status** | |
| Yes | 70(19.8) |
| No | 165(46.6) |
| Declined to answer | 119(33.6) |
| **Condom use non-regular partner in last 6 months**[*] | |
| YES all times | 62(32.6) |
| YES sometimes | 44(23.2) |
| Not at all | 77(40.5) |
| Decline to respond | 7(3.7) |
| **Condom use with the last 3 sex rounds** | |
| Yes all times | 48(25.2) |
| Yes sometimes | 60(31.6) |
| Not at all | 72(37.9) |
| Decline to respond | 10(5.3) |
| **Paid money or other variables to have sex** | |
| Yes | 128(67.4) |
| No | 59(31) |
| Declined to Answer | 3(1.6) |
| **Ever taken drugs before sex** | |
| Yes | 191(53.9) |
| No | 163(46.1) |
| **Had sex in exchange for drugs** | |
| Yes | 80(45.3) |
| No | 101(53.1) |
| Declined to answer | 3(1.6) |

[*]n=190, only those with more than one sexual partner, were assessed for consistent condom use in the last 6 months

months, and consistent condom use with non-regular partners were not statistically significant to an individual's HIV status (Pavlues>0.05). At multivariate analysis, females compared to the males were nearly four times more likely to report HIV-positive status (3.8, 95%CI: 1.01 - 15.5). Participants who reported having difficulty in accessing sterile injection materials were likely to be HIV positive, (4.69, 95%CI: 1.15 - 19.16). (Table 4).

## Discussion

In this study, seven in ten injection drug users were males, not married, unemployed, with higher levels of education, and of varied ages. Over nine in ten were Ugandans and seven in ten had ever been imprisoned. Peer influence was cited by seven in ten participants as a key determinant of drug use. Additionally, seven in ten had injected drugs in the last three months, with half reporting at least one mental health problem. One in three reported experiencing fever or cough in the last three months. Risky behaviour was common, with one in ten reporting syringe and needle sharing. A half had sex with non-regular partners, but only a third used condoms. Three-quarters engaged in transactional sex, yet only seven in ten knew

**Table 4. Association between socio-demographics, risk behaviours and HIV status of injection drug users in Kampala Uganda.**

| Variable | Unadjusted OR (95% CI) | P-Value | Adjusted OR (95% CI) | P-Value |
|---|---|---|---|---|
| **Sex** Male | 1.0 | | 1.0 | |
| Female | 4.67 (1.5-14.7) | 0.008 | 3.8 (0.9-15.5) | 0.049 |
| **Age** | | | | |
| 18-24yrs | 1.0 | | 1.0 | |
| 25-34yrs | 0.28 (0.07-1.18) | 0.082 | 0.36 (0.78-1.66) | 0.189 |
| 35yrs &above | 0.61 (0.16-2.38) | 0.48 | 1.02 (0.21-4.89) | 0.985 |
| **Nationality** | | | | |
| Ugandan | 1.0 | | | |
| Non-Ugandan | 6.52 (1.62-26.18) | 0.008 | 3.73 (0.69- 20.27) | 0.128 |
| **Access to sterile injection materials** | | | | |
| Easy | 1.0 | | 1.0 | |
| Not easy | 5.18 (1.56-17.18) | 0.007 | **4.69 (1.15-19.16)** | 0.031 |
| Decline to respond | 0.85 (0.10-7.06) | 0.878 | 0.83 (0.08-8.7) | 0.876 |
| **Sex with a non-regular partner** | | | | |
| Yes | 1.0 | | 1.0 | |
| No | 0.23 (0.05-1.10) | 0.067 | 0.51 (0.07-3.93) | 0.517 |
| decline to respond | 1.8 (0.21-15.48) | 0.592 | 2.6 (0.21-31.71) | 0.453 |
| **No of sexual partners in last 6 months** | | | | |
| >one Partner | 1.0 | | 1.0 | |
| None/one Partner | 0.61 (0.13-2.86) | 0.531 | 0.71 (0.09-5.43) | 0.746 |
| Decline to respond | 0.18 (0.02-1.40) | 0.1 | 2.27 (0.17-30.03) | 0.533 |
| **Consistent condom use in last three rounds** | | | | |
| YES all times | 1.0 | | | |
| YES sometimes | 0.75 (0.14-3.89) | 0.732 | | |
| Not at all | 0.51 (0.11-2.36) | 0.388 | | |
| decline to respond | 0.38 (0.74-1.95) | 0.248 | | |
| **Take drugs before sex** | | | | |
| Yes | 1.0 | | 1.0 | |
| No | 0.20 (0.04-0.93) | 0.04 | 0.51 (0.07-3.9) | 0.524 |

their HIV status, with a positivity rate of 3.7%, higher among females and non-Ugandan nationals. Females and those with difficulty accessing sterile injection materials were more likely to be HIV positive.

Our results are comparable to an earlier descriptive studies in Uganda, that found PWID to be predominatly male, not married, with some earning a living through sex work [21]. Similar findings have been reported in other regions, highlighting the vulnerability of unemployed and economically disadvantaged populations to injecting drug use [22, 23]. The UNODC resource guide on counselling in targeted intervention for injecting drug users, emphasizes the importance of understanding the characterisitics of IDU, for the successful management of opioid use disorder. Further noting that IDUs are often the most vulnerable, underprivileged subgroup of the community, mainly men in their productive years, not in gainful employment due to intoxication, and frequent absenteeism at work. Similarly, reports from low- and middle-income Countries observed that injecting drug users were primarily men, unmarried, Christian, and unemployed [24, 25]. Although the gender gap in drug use is narrowing, our findings suggest otherwise that men in Uganda and possibly similar resource-limited settings shoulder a large portion of the drug injection burden [26, 27]. Peer influence was prominently cited as a predisposing factor for drug use, likely associated with the younger participants, as over 70% were below 35 years old. Young people often influence or imitate each other to demonstrate a sense of belonging. Indeed, eight out of ten participants reported that either friends or relatives introduced them to drug use. Similarly, a study among university students aged below 25 years found that nine out of ten participants identified peer influence as the main determinant of psychoactive substance use [28].

## Mental health

We found a high self-reported occurrence of mental health disorders, coupled with limited access to treatment services. Nearly half of the participants reported a diagnosis of co-existing mental disorder, with only one in ten receiving treatment. Depression, Anxiety, and Attention Deficit Hyperactivity Disorder were the most prevalent co-morbidities. Schizo-phrenia, and post-traumatic stress disorder (PTSD). The prevalence of mental disorders in this study was high compared to what was found in a systematic review of 24 studies from Uganda, which reported a prevalence of 24.2%, in adults [29, 30]. In the same review, the combined prevalence of anxiety disorders and depressive disorders was approximately one in four adults. Globally, the prevalence of mental disorders has been found to vary widely based on the method of ascertainment, with recent studies reporting a prevalence of depressive disorders varying from as low as 3·4% to a high 66% in the general population [2]. Despite potential social desirability bias associated with self-reported data, our findings suggest that the escalating injecting drug use is likely exacerbating the ongoing mental health crisis in Uganda. Diagnosing mental disorders can be challenging, particularly in low-resource settings like Uganda where the inadequate human resources to address the growing mental disorder and drug use disorder burden are limited [31]. The signs and symptoms of acute intoxication and withdrawal syndrom are similar to those of other psychiatric disorders, complicating the accurate diagnosis of these conditions. Furthermore, mental disorders can be risk factors for and can exacerbate, substance use, and vice-versa, creating a complex interplay between the two [32]. This interplay highlights the need for integrated approaches to address both mental health disorders and opioid use disorders in Uganda.

## Physical health profiles

Injecting drug users face a significant burden of physical health issues, including infectious diseases such as malaria, sexually transmitted infections (STIs), and HIV. The reported

prevalence of malaria treatment (22%) is consistent with the high malaria burden in Uganda, as the leading cause of morbidity and mortality [33]. The prevalence of STIs (15%) underscores the intersection of substance use and sexual health risks among this population. These findings align with other studies, which have similarly reported elevated rates of STIs, HIV and other communicable diseases, among people who use drugs. The high vulnerability has been associated with some factors including high-risk sexual behaviors associated with drug use, limitations in accessing healthcare services, and hygiene practices, among others [34, 35]. The presence of tuberculosis (TB) and skin diseases among participants highlights the broader health challenges faced by PWID [27]. Overall, these findings emphasize the need for integrated healthcare interventions that address both infectious and non-communicable diseases among injecting users in Uganda and similar settings.

## HIV prevalence

The prevalence of HIV among injecting drug users was 3.7%, far less than the estimated 33% reported by the Uganda AIDS Commission and Uganda Ministry of Health [36]. However the recent crane survey and two other studies reported that the HIV prevalence among PWID in Uganda ranged between 3.6% to 17% [37–40]. Our earlier study reported an HIV-positive rate (8.8%) among injecting drug users enrolled in the Medication for Opioid Use Disorder (MOUD) program, where participants underwent routine HIV testing every three months.

The low prevalence may be attributed to the low frequency of injecting drug use in this population, which is largely due to the unaffordability of accessing injecting drugs and related paraphernalia[41]. Futher still, even with the low prevalence reported, the four times higher HIV Prevalence among females than males, is consistent with studies in Tanzania where the HIV-positive sero-status was three times higher in females compared to males [42]. Risky injection practices and sexual behaviour contribute to the heightened risk of HIV transmission among injecting drug users[27,43]. Sharing of injecting equipment and engaging in unprotected sexual encounters with non-regular partners were common. The high frequency of transactional sex and the use of drugs before sexual encounters further compound the risk of HIV transmission. The prevalent stigma contributing to low HIV status awareness (with only one in five injecting drug users aware of their status) also suggests inadequate coverage of HIV prevention services.38[43]. At the time of data collection, only one MOUD clinic existed in Kampala city. There remains urgent need to expand harm reduction services, including HIV testing, to reach more injecting drug users in Uganda.

## Risk factors for HIV infection

Although most participants were found to be occasional injectors, risky drug injection and sexual practices were common, posing a significant potential for transmitting and increasing the number of new HIV infections. Reported risky injection practices included sharing needles and syringes, sharing mixing equipment, flushing blood, and reusing syringes multiple times.These findings reinforce earlier reports from studies in Uganda [17,39]. Prevoius studies associated prevalent drug risk behavior with the unaffordability of new syringes and the nadir existence of harm reduction programs in Uganda. Regarding to risky sexual practices, over half of the participants reported having sex with a nonregular partner, three of ten reported inconsistent condom use, and over two-thirds of the participants were engaging in transactional sex. These factors heighten the vulnerability of PWID to HIV infection [44]. While taking drugs before sex was associated with an increased risk of HIV, this association was not significant in multivariate, similar to other sexual risk factors. A previous study by the Community Health Alliance Uganda (CHAU) examined discussed three main HIV risk

perspectives: impaired judgment leading to unprotected sex, frequent sexual encounters with non-regular partners, and reduced libido due to drug use, which lowers sexual activity[39]. Additionally, people who inject drugs may be forced to sell sex to acquire money to buy drugs [34,45]. In our study, nearly seven in ten of those who had multiple sexual encounters, paid money or other commodities in exchange of sex with another person, and nearly half reported having had sex in exchange for drugs.

Another crucial risk factor deserving attention is the status of being non-Ugandan, who were mostly refugees of recent immigration status. High proportions of non-Ugandans were found HIV positive but with no significant association in multivariate analysis. Studies conducted elsewhere in Dar-es-Salaam and the United States reported that recent immigrants had an increased risk of HIV infection [6,46]. This heightened risk among non-nationals may be attributed to various factors, including unawareness of safer needle use practices, financial constraints hindering the purchase of new needles, and exposure to circumstances that increase the likelihood of engaging in risky sex work.

The small sample size and missing data on key parameters may partially explain the failure to demonstrate an association between injection drug risk and the HIV status. Despite efforts to ensure the confidentiality and safety of participants during interviews, it was not uncommon for participants to decline to respond to particular essential questions or take an HIV test. Another factor that we didn't find significant was the frequency of injection; in Uganda, most drug users only injected occasionally. This affects the risk of exposure to HIV and programming for harm reduction interventions such as clean needle and syringe distribution. Nonetheless, our study finding reinforces current evidence on the relation between HIV serostatus, and injecting drug and sexual risk practices. Two studies conducted in Uganda found that participants reused syringes in some instances up to 2-4 times before disposal [17,47].

## Limitations

Selection bias may have emerged due to increased security enforcement in preparation for a Non-Aligned Movement (NAM) Summit in Kampala, during January 2024, as it coincided with the data collection period. This security nforcement drove more PWID into hiding, reducing our access to some community members. Selecting initial seeds on a first-come basis likely favoured accessible or motivated individuals, leading to an unrepresentative sample. This may underrepresent those more stigmatised, severely addicted, or with poor health and less access to services, skewing results and underestimating the true prevalence of risk behaviour and mental and physical health conditions.

In some instances, participants declined to answer or did not respond to a question. Although this could have been because individuals deliberately preferred not to answer, we can't rule out the fact that cognitive impairments, mood disorders, or other mental health issues suffered by people suffering from opioid use disorder, could compromise the ability to recall and report information accurately.

Relying on self-reporting by participants could have introduced recall bias, as participants might not accurately remember past events or might selectively report on such events, leading to misrepresentation of the true measures. To minimize this bias, the questions were designed with a maximum recall duration of three months, pretested, and simplified to reduce cognitive load and facilitate easier recall of past events.

Additionally, interviews were rescheduled based on participants' availability to enhance recall accuracy, avoid distractions, and allow for better concentration and memory recall. Furthermore, while measures were taken to ensure confidentiality and safety, the possibility of overreporting or underreporting drug and sexual risk behaviour remains, as participants may

have been influenced by social desirability bias stemming from internal stigma. The extended data collection period(six months) allowed us to reach and enroll injecting drug users of different categories. Collecting data at community venues where most injecting drug users congregate helped ensure diversity. Using ODK-enabled real-time data correction, increasing accuracy. Based on these adjustments, we believe these results are valid and representative of injecting drug users in Kampala.

## Conclusion

Our study provides valuable insight into the socio-demographic, mental, physical health, and HIV-associated risk of persons with drug user disorders in Kampala, Uganda. The findings indicate a significant vulnerability to injecting drug use, mental health disorders, and high-risk behaviour that predispose this population to HIV infection. Despite a low HIV prevalence compared to previous estimates, the interplay between drug use, risky injecting practices, and sexual behaviour suggests an urgent need for targeted interventions to address these intertwined challenges. Implementing a multi-sectoral approach that combines, harm reduction, physical healthcare, and mental health services is crucial for improving the well-being of persons who inject illicit drugs, and reducing the burden of HIV and other associated health issues in Uganda.

## Supporting information

**S1 Checklist.  Inclusivity in global research.**
(DOCX)

**S1 Data.  Dataset.**
(CSV)

## Acknowledgment

The authors gratefully acknowledge the support from the Butabika National Referral Mental Hospital management, Uganda Harm Reduction Network, and MOUD clinic staff who participated in the data collection and cleaning. Special thanks go to the Ugandan government and the United States PEPFAR Uganda Centers for Disease Control and Prevention (CDC) for supporting the establishment of the MOUD programme in the country.

## Author contributions

**Conceptualization:** Peter Mudiope, Bradley Mathers, Byamah Brian Mutamba, Stella Alamo, Fredrick Makumbi, Miriam Laker-Oketta, Rhoda Wanyenze.

**Data curation:** Peter Mudiope, Samuel Mutyaba.

**Formal analysis:** Peter Mudiope, Samuel Mutyaba, Nicholus Nanyeenya, Fredrick Makumbi.

**Funding acquisition:** Peter Mudiope, Joanita Nangendo.

**Investigation:** Peter Mudiope.

**Methodology:** Peter Mudiope, Stella Alamo, Fredrick Makumbi, Rhoda Wanyenze.

**Project administration:** Peter Mudiope, Byamah Brian Mutamba, Stella Alamo.

**Supervision:** Peter Mudiope, Joanita Nangendo, Byamah Brian Mutamba, Stella Alamo, Miriam Laker-Oketta, Rhoda Wanyenze.

**Validation:** Peter Mudiope, Bradley Mathers, Samuel Mutyaba, Fredrick Makumbi, Rhoda Wanyenze.

**Visualization:** Peter Mudiope, Samuel Mutyaba, Nicholus Nanyeenya.

**Writing – original draft:** Peter Mudiope, Bradley Mathers, Joanita Nangendo, Samuel Mutyaba, Stella Alamo, Nicholus Nanyeenya, Fredrick Makumbi, Miriam Laker-Oketta, Rhoda Wanyenze.

**Writing – review & editing:** Peter Mudiope, Bradley Mathers, Joanita Nangendo, Byamah Brian Mutamba, Nicholus Nanyeenya, Fredrick Makumbi, Miriam Laker-Oketta, Rhoda Wanyenze.

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
