## [Decision Letter · Decision Letter 0]

25 Sep 2024

PGPH-D-24-01264

Characterising People who inject drugs, and association with HIV infection: A Situation Analysis in Kampala city, Uganda

Dear Dr. Mudiope,

Thank you for submitting your manuscript to PLOS Global Public Health. After careful consideration, we feel that it has merit but does not fully meet PLOS Global Public Health’s publication criteria as it currently stands. Therefore, we invite you to submit a revised version of the manuscript that addresses the points raised during the review process.

You will see that the Reviewer has some substantial methodological concerns and I hope you will be able to revise your manuscript to address them.  I think that this is valuable data you have collected and I am optimistic that you will be able to explain what you have collected and how it should be interpreted in a way that will be comprehensible to readers like Reviewer 1.

We look forward to receiving your revised manuscript.

Kind regards,

Abraham D. Flaxman, Ph.D.

Academic Editor

Journal Requirements:

1. Please ensure you have stated the type of consent (Written) in the Methods section of your manuscript text to fully comply with the PLOS ONE policy on reporting research involving human participants. This information is currently provided only in the Human Participants Research Checklist, which will not be published with your manuscript files.

2. Please include a complete copy of PLOS’ questionnaire on inclusivity in global research in your revised manuscript. Our policy for research in this area aims to improve transparency in the reporting of research performed outside of researchers’ own country or community. The policy applies to researchers who have travelled to a different country to conduct research, research with Indigenous populations or their lands, and research on cultural artefacts. The questionnaire can also be requested at the journal’s discretion for any other submissions, even if these conditions are not met. Please find more information on the policy and a link to download a blank copy of the questionnaire here: https://journals.plos.org/globalpublichealth/s/best-practices-in-research-reporting. Please upload a completed version of your questionnaire as Supporting Information when you resubmit your manuscript.

3. In the online submission form, you indicated that [The datasets analysed during the current study will be shared on reasonable request by the corresponding]. 

a. In a public repository, 

b. Within the manuscript itself, or 

c. Uploaded as supplementary information.

4. Please provide separate figure files in .tif or .eps format.

Additional Editor Comments (if provided):

Reviewers' comments:

Reviewer's Responses to Questions

**Comments to the Author**

1. Does this manuscript meet PLOS Global Public Health’s publication criteria ? Is the manuscript technically sound, and do the data support the conclusions? The manuscript must describe methodologically and ethically rigorous research with conclusions that are appropriately drawn based on the data presented.

Reviewer #1: Partly

2. Has the statistical analysis been performed appropriately and rigorously?

Reviewer #1: No

3. Have the authors made all data underlying the findings in their manuscript fully available (please refer to the Data Availability Statement at the start of the manuscript PDF file)?

Reviewer #1: Yes

4. Is the manuscript presented in an intelligible fashion and written in standard English?

Reviewer #1: No

5. Review Comments to the Author

Reviewer #1: This cross-sectional study examined HIV prevalence and other sociodemographic and health-related characteristics of people who inject drugs (PWID) in Kampala, Uganda in 2023. While focuses on an important topic, there are a few issues that the manuscript would benefit from addressing.

Major comments:

1. HIV testing: There are two places in the manuscript that described the way HIV testing was done (“study measurements” section in the methods and “HIV prevalence” section in discussion), and they describe it in very different ways. This inconsistency needs to be resolved. Importantly, in the discussion section, they indicated that participants who self-reported a recent negative test (past 3 or 6 months? Timeframe is inconsistently written, too) did not undergo HIV rapid testing through the study. This is not a reliable way to estimate HIV prevalence. And indeed, the study found quite low HIV prevalence. With the ambiguity of HIV testing method, I am not sure about the validity of this finding.

2. Frequency of injection drug use, types of drugs injected and non-injection drug use: In “Drug use-related characteristics” section in the results, they report that 50% of their sample last injected drugs more than a month before the interview. This suggests that study participants are mostly very occasional injectors, and is not aligned with conventional characteristics of PWID discussed in the context of HIV/AIDS (i.e., usually people inject drugs on a much more frequent basis). Despite this, the authors do not discuss this point in the discussion section – is it common in Uganda that people inject drugs so infrequently or is it just that the study did not reach those who engage in more intense injection drug use patterns?

Also, they do not report any types of drugs injected or non-injection drug use patterns. Without such information, they cannot claim that they are providing “a comprehensive insight” into the characteristics of PWID in Kampala. The low frequency of injection drug use may mean that those participants were engaged more in non-injection drug use.

3. Multivariable regression modeling procedure (statistical analysis section in the methods) is unclear. They stated that they included “variables that had p values <0.1 or known association with HIV serostatus, were selected for inclusion.” Which variables are the known association with HIV serostatus? Were variables that were not hypothesized to be associated with HIV serostatus but had p values <0.1 still included in the multivariable model? They should not.

4. In “Risk factors for HIV injection” in the discussion, the authors stated that: “The common risk drug injection practices reported include sharing needles and syringes, sharing mixing equipment, flushing blood, and reusing syringes multiple times.” But the prevalence of these behaviours was less than 10%. I don’t think we call it “common” behaviour. Again, this raises a question about who the study sample was – weren’t they mostly drug smokers, not injectors?

5. “HIV Risk behavior” and “Drug use and sexual-related HIV risk factors” sections in the results include some interpretation of the results (e.g., “…was a significant concern”). Such interpretation should be removed from the results section and placed in the discussion section.

6. “Risk factors for HIV injection” in the discussion section includes a discussion of non-nationals, and the authors stated that: “The risk of being HIV positive was found to be

five times higher among non-nationals compared to Ugandans.” As the regression results did not show a statistically significant association and the 95% CI is very wide, this statement as well as the related discussion should be removed.

Minor comments

7. Introduction, 1st para: “The rising burden of drug use including injecting, continues to harm the economic, health, and social sectors(1).” This sentence is misleading and should be revised. It is not the drug use per se that harms the society, but the way drug use is regulated and viewed in the society.

8. “Study population, sampling and sample size” and “Study procedure” sections in the methods can be merged. The two sections include descriptions of how the researchers recruited participants.

9. Abstract, methods: The authors stated that they used “a semi-structured questionnaire”; however, it should be a structured questionnaire. A semi-structured questionnaire is usually used for qualitative interviews.

10. “Mental health” section in the discussion: The two sentences are conflicting as ADHD appears in both sentences. This should be corrected. “Depression, Anxiety, and Attention Deficit Hyperactivity Disorder were the most prevalent co-morbidities. Schizophrenia, post-traumatic stress disorder (PTSD), and attention deficit hyperactivity

disorder (ADHD) were less common in this population.”

11. Overall, English writing and typos should be checked and corrected.

6. PLOS authors have the option to publish the peer review history of their article (what does this mean? ). If published, this will include your full peer review and any attached files.

**Do you want your identity to be public for this peer review?** For information about this choice, including consent withdrawal, please see our Privacy Policy .

Reviewer #1: No

---

## [Decision Letter · Decision Letter 1]

15 Dec 2024

PGPH-D-24-01264R1

Characterising People who inject drugs, and association with HIV infection: A Situation Analysis in Kampala city, Uganda

Dear Dr. Mudiope,

Thank you for submitting your manuscript to PLOS Global Public Health. After careful consideration, we feel that it has merit but does not fully meet PLOS Global Public Health’s publication criteria as it currently stands. Therefore, we invite you to submit a revised version of the manuscript that addresses the points raised during the review process.

Please give this a careful edit, in light of the inconsistencies the reviewers have identified, and be sure to incorporate the suggestions from Reviewer 2 to make sure your readers are not distracted by your terminology.

We look forward to receiving your revised manuscript.

Kind regards,

Abraham D. Flaxman, Ph.D.

Academic Editor

Journal Requirements:

Additional Editor Comments (if provided):

Reviewers' comments:

Reviewer's Responses to Questions

**Comments to the Author**

1. If the authors have adequately addressed your comments raised in a previous round of review and you feel that this manuscript is now acceptable for publication, you may indicate that here to bypass the “Comments to the Author” section, enter your conflict of interest statement in the “Confidential to Editor” section, and submit your "Accept" recommendation.

Reviewer #1: (No Response)

Reviewer #2: All comments have been addressed

2. Does this manuscript meet PLOS Global Public Health’s publication criteria ? Is the manuscript technically sound, and do the data support the conclusions? The manuscript must describe methodologically and ethically rigorous research with conclusions that are appropriately drawn based on the data presented.

Reviewer #1: Yes

Reviewer #2: Yes

3. Has the statistical analysis been performed appropriately and rigorously?

Reviewer #1: Yes

Reviewer #2: I don't know

4. Have the authors made all data underlying the findings in their manuscript fully available (please refer to the Data Availability Statement at the start of the manuscript PDF file)?

Reviewer #1: (No Response)

Reviewer #2: Yes

5. Is the manuscript presented in an intelligible fashion and written in standard English?

Reviewer #1: Yes

Reviewer #2: No

6. Review Comments to the Author

Reviewer #1: The authors addressed most of my previous comments. I have a few minor comments.

1) The sample size seems to have been reduced from 499 to 354. However, there is no explanation for it. Why has it been reduced?

2) Abstract, Conclusion, and the Conclusion section in the manuscript: "Our study provides a comprehensive insight into..." I still think that the term "comprehensive" is a stretch. I suggest that the authors tone it down.

3) Introduction: "the rising burden of illicit drug use including injecting, continues to harm the economic, health, and social sectors." Again, I would argue that the current ways in which (illicit) drug use is regulated are causing the harm, not the (illicit) drug use per se. I defer to the Editor on this point.

4) Results, "HIV risk behavior" and "Drug use and sexual-related HIV risk factors" sections: These sections still contain some interpretation of the results. I defer to the Editor on this point.

5) Discussion, HIV prevalence: It is reported as 4.38%. However the results section reports it as 3.7%. The discrepancy needs to be resolved.

Reviewer #2: Thank you for this revised version. I would encourage you to reflect further on the comments and ensure that the abstract and paper - throughout - are coherent. As an example, you note in the limitations, there were some issues with gaining access to the population for this study, so it is not correct to claim this provides comprehensive insights.

I would encourage you to be very careful with the use of language - `HIV positives' is not acceptable since it defines a person by an infection. Please follow the People First Charter language. Likewise, the term `developing countries' was replaced many years ago by other terms.

Page 4 - rather than using the word `hotspots' describe the locations. What types of place did you meet people?

Page 5 and elsewhere - do you mean `flash blood syringes' or `flashblood'? And I suggest you explain what this is - some readers may not know.

Page 5 - was clearance gained from UNCST and CDC in addition to the Makerere REC?

Page 6 - Roman Catholic - consider use of capital letters for religious denominations/faiths since you have now used that for Anglican.

The paper requires very careful language editing. There are inconsistences - for example, data were/data was (use date were throughout). I suggest you employ an English language editor to help you.

7. PLOS authors have the option to publish the peer review history of their article (what does this mean? ). If published, this will include your full peer review and any attached files.

**Do you want your identity to be public for this peer review?** For information about this choice, including consent withdrawal, please see our Privacy Policy .

Reviewer #1: No

Reviewer #2: No

---

## [Decision Letter · Decision Letter 2]

29 Jan 2025

Characterising People who inject drugs, and association with HIV infection: A Situation Analysis in Kampala city, Uganda

PGPH-D-24-01264R2

Dear Dr Mudiope,

We are pleased to inform you that your manuscript 'Characterising People who inject drugs, and association with HIV infection: A Situation Analysis in Kampala city, Uganda' has been provisionally accepted for publication in PLOS Global Public Health.

Best regards,

Abraham D. Flaxman, Ph.D.

Academic Editor

Please proofread carefully, it would be a shame for this valuable work to get let traction than it deserves because of typos.

Reviewer Comments (if any, and for reference):

Reviewer's Responses to Questions

**Comments to the Author**

1. If the authors have adequately addressed your comments raised in a previous round of review and you feel that this manuscript is now acceptable for publication, you may indicate that here to bypass the “Comments to the Author” section, enter your conflict of interest statement in the “Confidential to Editor” section, and submit your "Accept" recommendation.

Reviewer #1: (No Response)

Reviewer #2: All comments have been addressed

2. Does this manuscript meet PLOS Global Public Health’s publication criteria ? Is the manuscript technically sound, and do the data support the conclusions? The manuscript must describe methodologically and ethically rigorous research with conclusions that are appropriately drawn based on the data presented.

Reviewer #1: (No Response)

Reviewer #2: Yes

3. Has the statistical analysis been performed appropriately and rigorously?

Reviewer #1: (No Response)

Reviewer #2: N/A

4. Have the authors made all data underlying the findings in their manuscript fully available (please refer to the Data Availability Statement at the start of the manuscript PDF file)?

Reviewer #1: (No Response)

Reviewer #2: Yes

5. Is the manuscript presented in an intelligible fashion and written in standard English?

Reviewer #1: (No Response)

Reviewer #2: Yes

6. Review Comments to the Author

Reviewer #1: The authors addressed most of my comments. Although, the added phrase to the Introduction section "challenges related to its regulation" is unclear. Causes of harm should be more clearly articulated (e.g., criminalization). I will defer to the Editor on this point.

Reviewer #2: Thank you for addressing my comments - no further comments.

7. PLOS authors have the option to publish the peer review history of their article (what does this mean? ). If published, this will include your full peer review and any attached files.

**Do you want your identity to be public for this peer review?** For information about this choice, including consent withdrawal, please see our Privacy Policy .

Reviewer #1: No

Reviewer #2: No
